

# Bioinformatics analysis and identification of hub genes and immune-related molecular mechanisms in chronic myeloid leukemia

Fangyi Yao, Cui Zhao, Fangmin Zhong, Tingyu Qin, Shuqi Li, Jing Liu, Bo Huang and Xiaozhong Wang

Jiangxi Province Key Laboratory of Laboratory Medicine, Department of Clinical Laboratory, the Second Affiliated Hospital of Nanchang University, Nanchang, Jiangxi, China

Corresponding author
Xiaozhong Wang,
wangxiaozhong@ncu.edu.cn

## ABSTRACT

**Background:** Chronic myeloid leukemia (CML) is a malignant hyperplastic tumor of the bone marrow originating from pluripotent hematopoietic stem cells. The advent of tyrosine kinase inhibitors (TKIs) has greatly improved the survival rate of patients with CML. However, TKI-resistance leads to the disease recurrence and progression. This study aimed to identify immune-related genes (IRGs) associated with CML progression.

**Methods:** We extracted the gene's expression profiles from the Gene Expression Omnibus (GEO). Bioinformatics analysis was used to determine the differentially expressed IRGs of CML and normal peripheral blood mononuclear cells (PBMCs). Functional enrichment and gene set enrichment analysis (GSEA) were used to explore its potential mechanism. Hub genes were identified using Molecular Complex Detection (MCODE) and the CytoHubba plugin. The hub genes' diagnostic value was evaluated using the receiver operating characteristic (ROC). The relative proportions of infiltrating immune cells in each CML sample were evaluated using CIBERSORT. Quantitative real-time PCR (RT-qPCR) was used to validate the hub gene expression in clinical samples.

**Results:** A total of 31 differentially expressed IRGs were identified. GO analyses revealed that the modules were typically enriched in the receptor ligand activity, cytokine activity, and endopeptidase activity. KEGG enrichment analysis of IRGs revealed that CML involved Th17 cell differentiation, the NF-kappa B signaling pathway, and cytokine-cytokine receptor interaction. A total of 10 hub genes were selected using the PPI network. GSEA showed that these hub genes were related to the gamma-interferon immune response, inflammatory response, and allograft rejection. ROC curve analysis suggested that six hub genes may be potential biomarkers for CML diagnosis. Further analysis indicated that immune cells were associated with the pathogenesis of CML. The RT-qPCR results showed that proteinase 3 (PRTN3), cathepsin G (CTSG), matrix metalloproteinase 9 (MMP9), resistin (RETN), eosinophil derived neurotoxin (RNase2), eosinophil cationic protein (ECP, RNase3) were significantly elevated in CML patients' PBMCs compared with healthy controls.

**Conclusion:** These results improved our understanding of the functional characteristics and immune-related molecular mechanisms involved in CML progression and provided potential diagnostic biomarkers and therapeutic targets.

## INTRODUCTION

Chronic myeloid leukemia (CML) is a malignant, clonal, and proliferative disease originating from hematopoietic stem cells (*Nash, 1999*). CML is characterized by the presence of a BCR-ABL1 fusion gene. This disease is classified into a chronic phase (CP), accelerated phase (AP), and a blast phase (BP). Tyrosine kinase inhibitors (TKIs), especially Imatinib (IM), have been used to treat CML with favorable outcomes (*Lugo et al., 1990*). However, TKI use may lead to side effects and drug resistance. This in turn may decrease TKI efficacy in some patients (*Hehlmann, 2012*). Thus, it is important to understand the mechanisms underlying CML and determine potential biomarkers.

Studies have proposed that gene expression-based characteristics may be used in CML diagnosis and prognosis (*Vinhas et al., 2017*), but they have not been made a part of routine clinical care due to insufficient validation cohorts. A number of new biomarkers have emerged following the validation of large-scale gene expression datasets (*Feng et al., 2020*). The establishment of multiple gene expression profiles makes it possible to identify reliable CML biomarkers. New evidence has suggested that the immune system, consisting of immune-related genes (IRGs) and tumor-infiltrating immune cells, is vital during cancer initiation and progression (*Zou & Hu, 2020*). A number of recent studies have constructed prognostic immune signals for use in the diagnosis and prognosis of a variety of cancers (*Geng et al., 2021*). *Hu et al. (2021)* constructed a seven-IRG signature using data from the Gene Expression Omnibus (GEO) database to predict glioblastoma multiform prognosis. This signature also revealed the potential functions of these genes in glioblastoma occurrence and development (*Hu et al., 2021*). (*Zhu et al. (2020)* predicted the survival rate of patients with acute myeloid leukemia with a normal karyotype by constructing a six-IRG signature to determine their prognostic characteristics. However, the clinical role of IRGs in CML has yet to be determined.

In this study, we identified IRGs using the data from the GEO database and performed network analyses to assess the molecular mechanisms underlying CML progression. Further, the biological processes involved were analyzed using gene ontology (GO) enrichment and kyoto encyclopedia of genes and genomes (KEGG) pathways for the differentially expressed genes. Moreover, the top 10 hub genes screened *via* protein-protein interaction (PPI) network were selected for their functional similarity, and their diagnostic value was assessed. Our results established the first comprehensive network of IRGs and provided a useful framework for explaining CML's molecular mechanisms at the systems biology level.

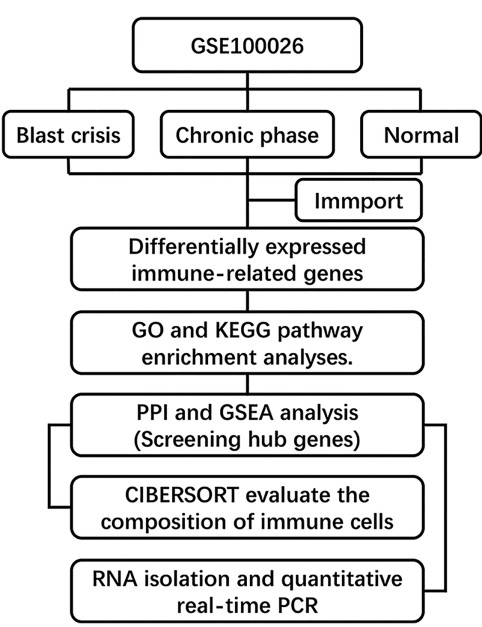

**Figure 1 Flow chart of methodologies applied in the study.**

# MATERIALS AND METHODS

## Data collection

The gene expression dataset GSE100026 (*Li et al., 2020*) was downloaded from the GEO database (GPL18573 Illumina NextSeq 500) and annotated using R software equipped with annotation files. This dataset included five CML chronic-phase (CML-CP) peripheral blood mononuclear cell (PBMCs) samples, five CML blast-period (CML-BP) PBMCs samples, and five normal PBMCs samples. We acquired 2,498 genes from the ImmPort database (https://immport.niaid.nih.gov) (*Bhattacharya et al., 2018*). Hallmark gene sets were extracted from the MSigDB database (http://software.broadinstitute.org/gsea/ msigdb/genesets.jsp). The overview of the workflow is shown in Fig. 1.

## Identifying IRGs

We converted the RNA sequencing (RNA-seq) data from RNA-Seq by Expectation Maximization (RSEM) into transcripts per kilobase million (TPM) expression profiles. Data analysis was performed using the limma package in R (*Ritchie et al., 2015*).
The cutoffs were set as $|\log FC| > 1$, $P < 0.05$, and the false discovery rate (FDR) < 0.05 to screen for differentially expressed genes. The intersection of the differentially expressed genes and the immune genes was used to find the IRGs. These results were displayed on a volcano map and a heat map (*Xue et al., 2020*).

## PPI network construction and module analysis

We used the Metascape database (https://metascape.org/gp/index.html#/main/step1) to perform GO and KEGG pathway enrichment analyses. We identified the most enriched biological pathways and functions related to the IRGs (*Zhou et al., 2019*). We then

constructed a protein-protein interaction (PPI) network of IRGs using the Search Tool for the Retrieval of Interacting Genes (STRING) online database (http://string-db.org, version 10). The network was visualized in Cytoscape (version 3.6.1) software (*Szklarczyk et al., 2017*). We used the Molecular Complex Detection (MCODE) and cytoHubba plug-ins to identify the merged network's hub genes. These genes serve as diagnostic markers (*Chin et al., 2014*).

## Verifying diagnostic markers

We performed receiver operating characteristic (ROC) curve analysis on each screened diagnostic marker in the GSE100026 dataset to verify its accuracy. R's pROC package was used for ROC curve analysis. Regarding the interpretation of AUC results, a test with an area >0.9 indicates high accuracy, 0.7–0.9 as moderate accuracy, 0.5–0.7 as low accuracy, and 0.5 as a chance result (*Akobeng, 2007*). Genes with an AUC value greater than 0.7 were retained.

## Gene set enrichment analysis (GSEA)

GSEA is a statistical method used to determine whether genes from a particular pathway or other predefined gene sets are differentially expressed in different phenotypes. Reactome pathways were analyzed with GSEA using clusterProfiler to define each functional cluster. C2.all.v6.2.symbols.gmt was selected as the reference gene set. A false discovery rate <0.1 and *P*-value < 0.01 were set as the cut-off criteria.

## Estimating immune cell type fractions

CIBERSORT is a bioinformatics method used to evaluate the composition of immune cells through standardized gene expression transformation (*Chen et al., 2018*). The CIBERSORT algorithm (CIBERSORT R script v1.03; http://cibersort.stanford.edu/) was used to analyze the immune infiltration of the CML-CP-Normal, CML-BP-Normal, and CML-CP-CML-BP datasets, respectively. This analysis helped us assess differences in immune cell infiltration among CML-CP, CML-BP, and healthy samples.

## Analysis of the relationship between hub genes and immune cells

The relationship between the six hub genes and the infiltrating immune cells was analyzed in three datasets. Hub genes expression levels were divided into low- or high-expression groups based on the genes' median expression levels. The Wilcoxon rank-sum test was used to analyze the difference of immune cell infiltration between the high and low levels of hub genes. Box plots were used to visualize the difference in infiltrating immune cell expression between both groups.

## Sample collection

Peripheral blood samples were collected from patients in the Second Affiliated Hospital of Nanchang University from March 2020 to February 2021. CML patients were pathologically diagnosed and categorized into CML-CP ($n = 25$) and CML-BP ($n = 25$) groups. The normal samples ($n = 25$) were obtained from healthy individuals. The characteristics of CML patients and healthy donors are presented in Table 1. PBMCs

**Table 1 Clinical characteristics.**

| Characteristic | CML-CP (*n* = 25) | CML-BP (*n* = 25) | Normal (*n* = 25) |
|---|---|---|---|
| Age (years), median (range) | 49 (19–69) | 55 (30–73) | 52 (21–70) |
| Male/female (n/n) | 13/12 | 15/10 | 14/11 |
| WBC count, ×$10^9$/L, median (range) | 138.3 (14.8–440.5) | 72.6 (2.5–381.9) | 6.4 (4.8–9.7) |
| Haemoglobin level (g/L) | 102 (50–143) | 79 (43–135) | 139 (120–156) |
| Platelet count, ×$10^9$/L, median (range) | 395 (6–1,309) | 50 (10–183) | 223 (105–318) |

Note:
BP, blast phase; CML, chronic myeloid leukemia; CP, chronic phase; WBC, white blood cells.

**Table 2 RT-qPCR primers.**

| Primers | Sequences (5′-3′) |
|---|---|
| GAPDH | Forward: ATGGTGAAGGTCGGTGTGAA |
| | Reverse: GAGTGGAGTCATACTGGAAC |
| CTSG | Forward: GAGTCAGACGGAATCGAAACG |
| | Reverse: CGGAGTGTATCTGTTCCCCTC |
| MMP9 | Forward: CGCAGACATCGTCATCCAGT |
| | Reverse: CGCAGACATCGTCATCCAGT |
| PRTN3 | Forward: CCCTGATCCACCCGAGATTC |
| | Reverse: GGTTCTCCTCGGGGGTTGTAA |
| RETN | Forward: GTGTGCCGGATTTGGTTAGC |
| | Reverse: GAGGGAACCAAGAGACCCAC |
| RNASE2 | Forward: ATCAACGACGAGACCCTCCA |
| | Reverse: AGGAGCTTGGCAGATGAGTG |
| RNASE3 | Forward: GATCCACGGGATTCTCCACG |
| | Reverse: GGAGCTTGGCAGATGAGTGA |

were isolated by density gradient centrifugation on a Ficoll-Paque (Sigma, USA) according to the manufacturer's protocol. Our study was approved by the Ethics Committee of the Second Affiliated Hospital of Nanchang University (approval no. 2017096) and all aspects of the study followed the guidelines set forth by the Helsinki Declaration. All participants provided signed informed consent.

### RNA isolation and quantitative real-time PCR (RT-qPCR)

Total RNA was extracted using TRIzol reagent (Invitrogen). Reverse transcription was performed using a PrimeScript RT Reagent Kit (TaKaRa, Dalian, China). We performed RT-qPCR using SYBR Premix Ex Taq (TaKaRa) on a ABI 7500 Real-Time PCR System (Applied Biosystems, Foster City, CA, USA). All primers were designed and synthesized by Nanjing Genscript Biotech Co., Ltd. GAPDH was used as the internal reference. The primers used in RT-qPCR assay are listed in Table 2. Indicated genes' fold changes were calculated using the $2^{-\Delta\Delta Ct}$ method.

## Statistical analysis

Statistical analysis was performed with R Software (version 3.6.2). A $P$-value < 0.05 was considered to be statistically significant.

## RESULTS

### Identification of IRGs

We used the limma package to analyze the differences between the CML-CP-Normal (CML-CP = 5, Normal = 5), CML-BP-Normal (CML-BP = 5, Normal = 5), and CML-CP-CML-BP (CML-CP = 5, CML-BP = 5) datasets. We ascertained where the differential genes and IRGs in each dataset intersected. From the three datasets, we identified 336, 159, and 376 differential genes, respectively (Figs. 2G–2I). CML-CP-Normal's clustering heat map is shown in Fig. 2A and its volcano map is shown in Fig. 2D. CML-BP-Normal's clustering heat map is shown in Fig. 2B and its volcano map is shown in Fig. 2E. CML-CP-CML-BP's clustering heat map is shown in Fig. 2C and its volcano map is shown in Fig. 2F. Thirty-one IRGs were obtained from the intersection of the differential genes and the immune genes in the three datasets (Fig. 2J).

### Functional enrichment analysis of IRGs

Thirty-one IRGs were enriched and analyzed using the Metascape database. The top GO terms related to biological processes (BC) were neutrophil activation, neutrophil degranulation, and neutrophil activation involved in immune response (Fig. 3A). The top GO terms related to cellular components (CC) included secretory granule lumen, cytoplasmic vesicle lumen, and vesicle lumen. The top GO terms related to molecular functions (MF) included receptor ligand activity, cytokine activity, and endopeptidase activity. KEGG analysis indicated that these IRGs were mainly enriched in Th17 cell differentiation, NF-kappa B signaling pathway, and cytokine-cytokine receptor interaction (Fig. 3B).

### PPI network analysis of IRGs

STRING was used to generate the PPI network, and the Cytoscape tool was used to visualize interactions between 31 IRGs (Figs. 4A and 4B). The cytoHubba and MCODE plug-ins were used to analyze the hub genes with maximum correlation criterion (MCC), and genes with the top 10 scores were identified as hub genes (Figs. 4C and 4D). Among these genes, *AZU1*, *CAMP*, *CCL5*, *CTSG*, *MMP9*, *MPO*, *PRTN3*, *RETN*, *RNASE2*, and *RNASE3* showed the highest node scores, suggesting that they may play causative roles in CML progression.

### GSEA analysis of hub genes

GSEA were performed on the 10 hub genes obtained from the GSE10002 dataset. The median of each gene expression value was used to categorize the patients into high or low groups. The msigDB database's hallmark gene set was used as a reference in our analysis. We identified different significant pathways between the higher- and lower-gene-expression groups. $P < 0.05$ and $|NES| > 1.0$ was considered to be significant. As shown in

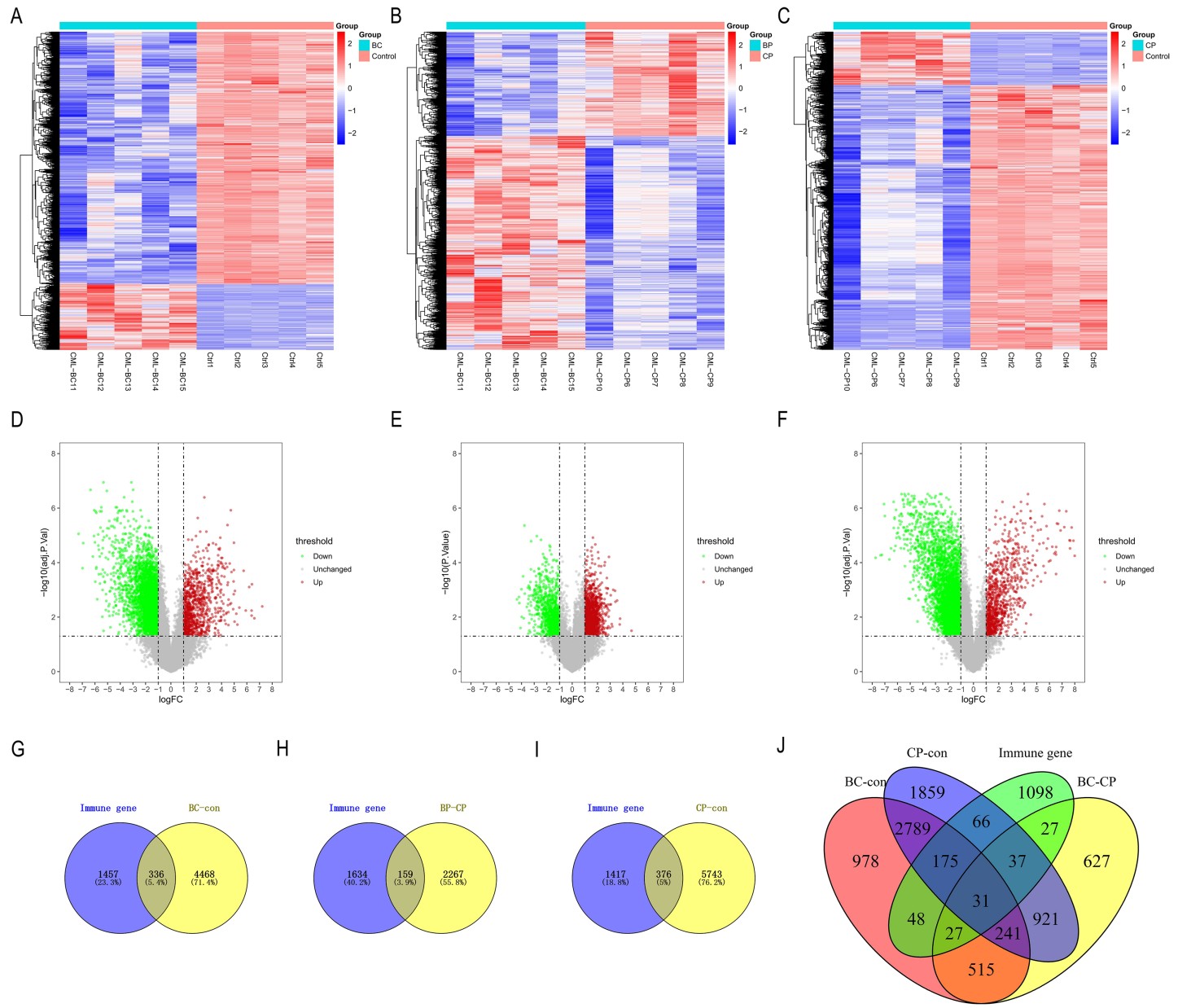

**Figure 2 Identification of IRGs in CML.** (A–C) Difference analysis between CML-CP-normal, CML-BP-normal and CML-CP-CML-BP cluster heat map. (D–F) Difference analysis between CML-CP-normal, CML-BP-normal and CML-CP-CML-BP volcano map. (G–I) CML-CP-normal, CML-BP-normal, CML-CP-CML-BP and immune gene intersection Venn diagram. (J) Intersection Venn diagram of the IRGs significantly associated with CML which were short-listed for the cross-validation.

Fig. 5, most pathways enriched in the hub genes high-expression cohort were immune-related, including interferon gamma response, IL2-STAT5 signaling pathway, complement, allograft rejection.

## Hub genes for CML diagnosis

To explore the accuracy of the top 10 hub genes as the diagnostic biomarkers for CML, the ROC curves were plotted, respectively (Fig. 6). Six hub genes with an AUC value greater

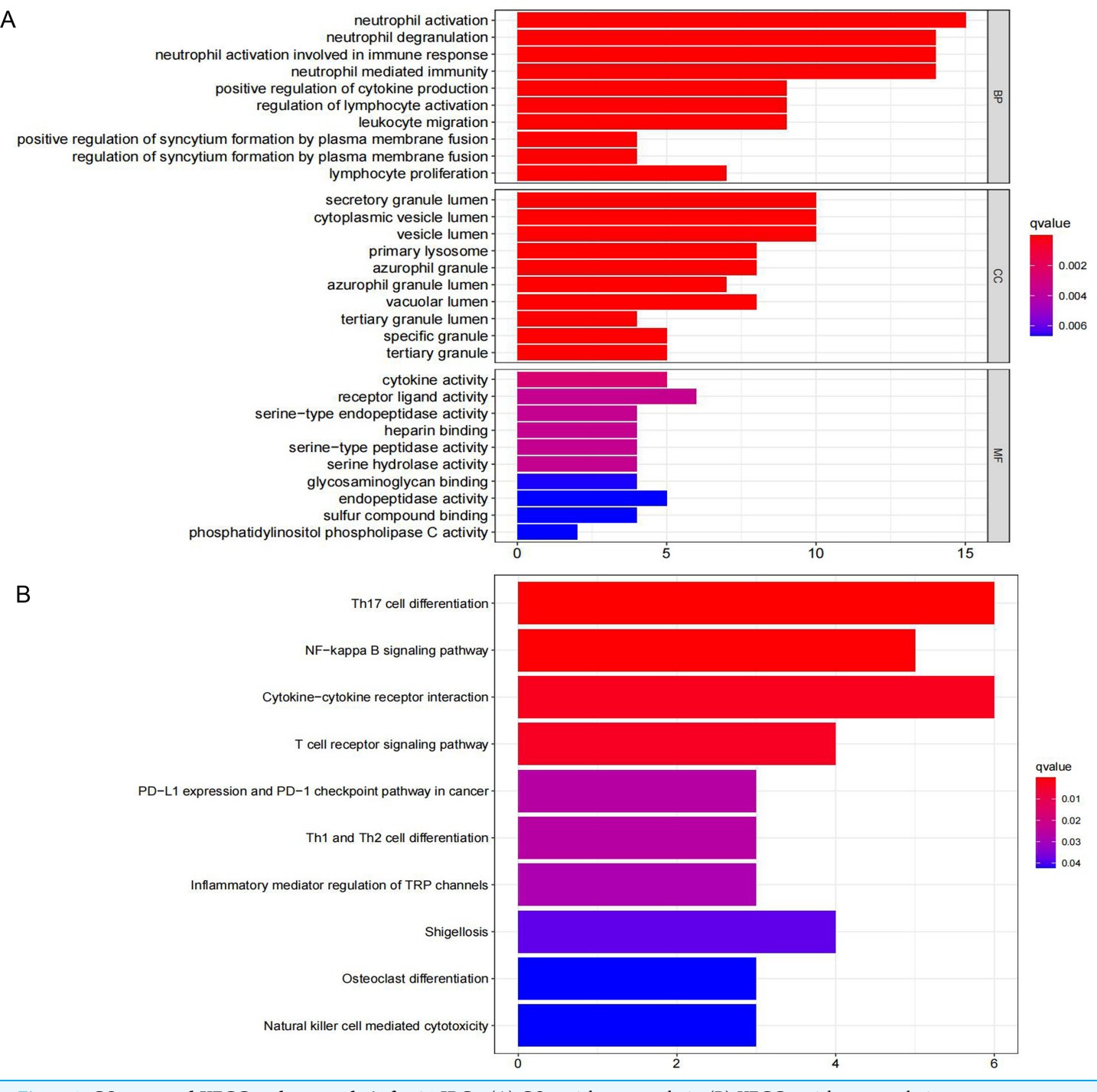

**Figure 3 GO term and KEGG pathway analysis for 31 IRGs.** (A) GO enrichment analysis. (B) KEGG enrichment analysis.

than 0.7 were used as diagnostic markers. Notably, the AUC for *CTSG* reached 0.96, and was the largest among the 10 hub genes. The other AUC values were 0.82, 0.74, 0.88, 0.84, and 0.88 for *MMP9*, *PRTN3*, *RETN*, *RNASE2*, and *RNASE3*, respectively. These results suggest that the 6 hub genes may serve as diagnostic biomarkers for CML.

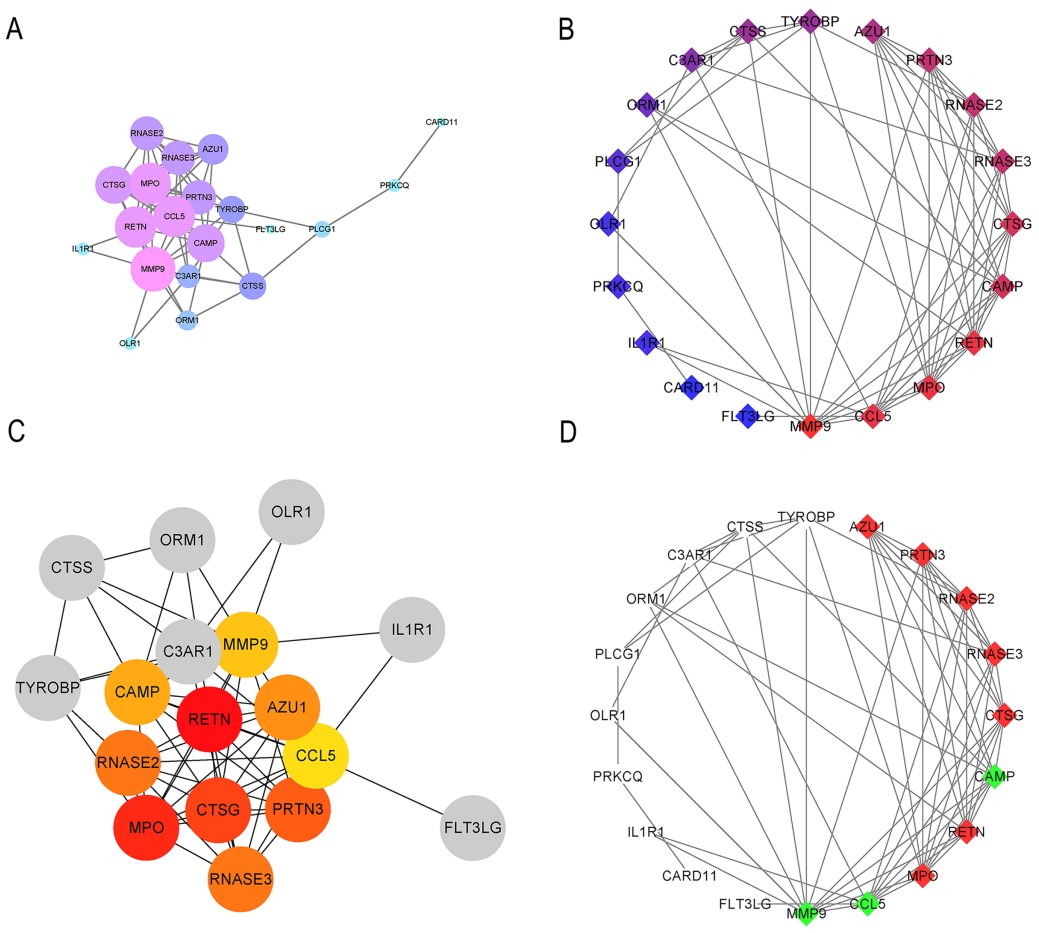

**Figure 4 Significant modular analysis based on PPI network.** (A–B) PPI network was constructed using a total of 31 IRGs. (C) Ten hub genes recognized by cytoHubba plug-in. (D) 10 hub gene recognized by MCODE plug-in.                         

## Immune infiltration analysis

We performed immune infiltration analysis on three datasets (Figs. 7–9). The distribution of the 22 types of infiltrating immune cells in each sample showed there to be immunological differences between the two groups (Figs. 7A, 8A, 9A). The abundance of immune cells in each sample is illustrated as a heatmap in Fig. 7B, 8B, 9B; the correlation between each of the infiltrated immune cell types is shown in Figs. 7C, 8C, 9C; and the violin plot in Fig. 7D, 8D, 9D visualizes the differences in each type of immune cell between the two groups.

## Relationships between the hub genes and immune cells

We analyzed the relationship between hub genes *CTSG*, *MMP9*, *PRTN3*, *RETN*, *RNASE2* and *RNASE3* and immune cells in the three datasets, respectively (Figs. 10–12). In the CML-BP-Normal datasets, *CTSG*, *MMP9*, *PRTN3*, *RNASE2* and *RNASE3* were significantly associated with the infiltration of monocytes, mast cells resting and NK cells (Fig. 10). In the CML-CP-CML-BP datasets, the six hub genes were significantly associated

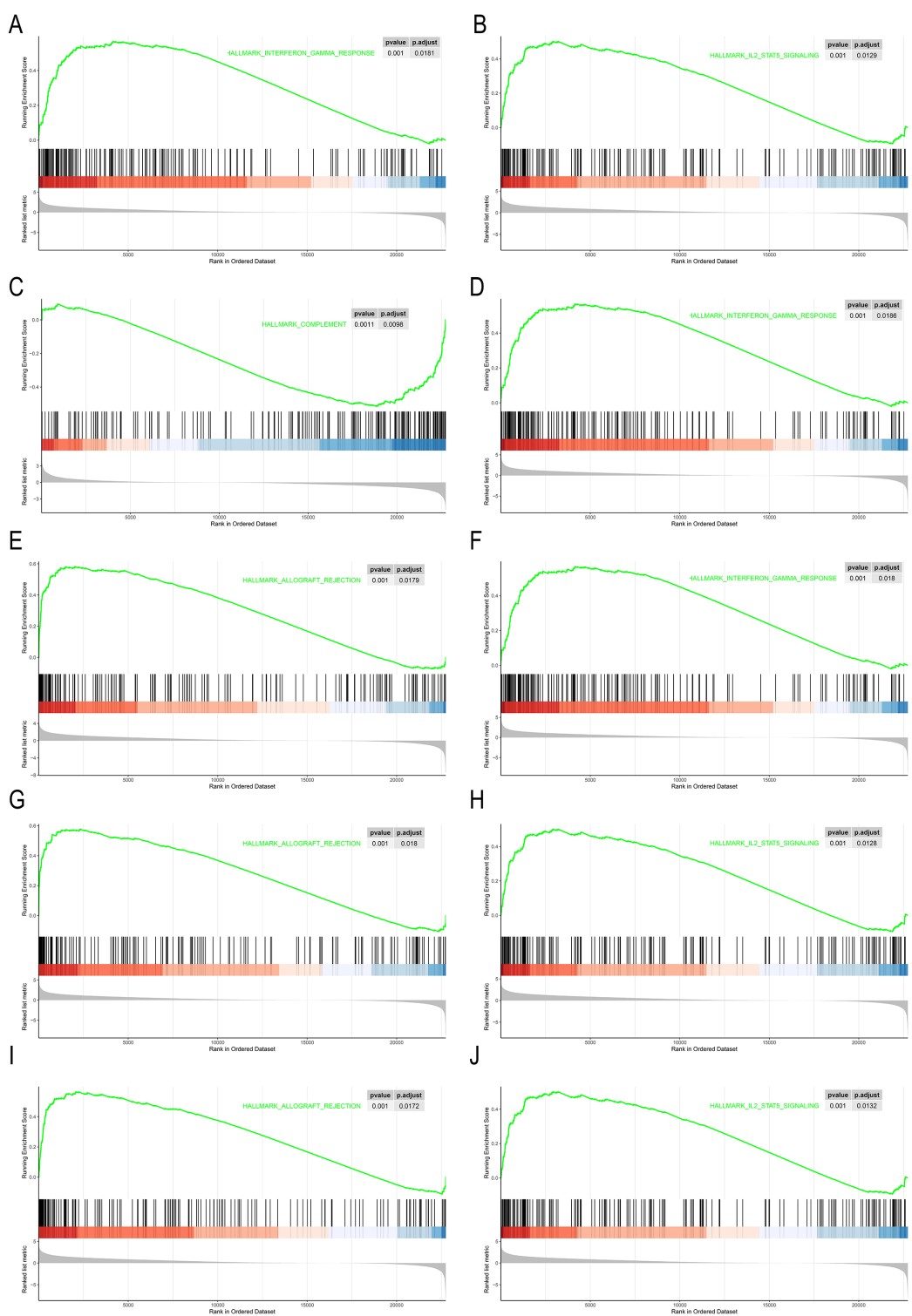

**Figure 5  GSEA of the 10 hub gene expression profiles of the hallmark gene set.** (A) AZU1. (B) CAMP. (C) CCL5. (D) CTSG. (E) MMP9. (F) MPO. (G) PRTN3. (H) RETN. (I) RNASE2. (J) RNASE3.

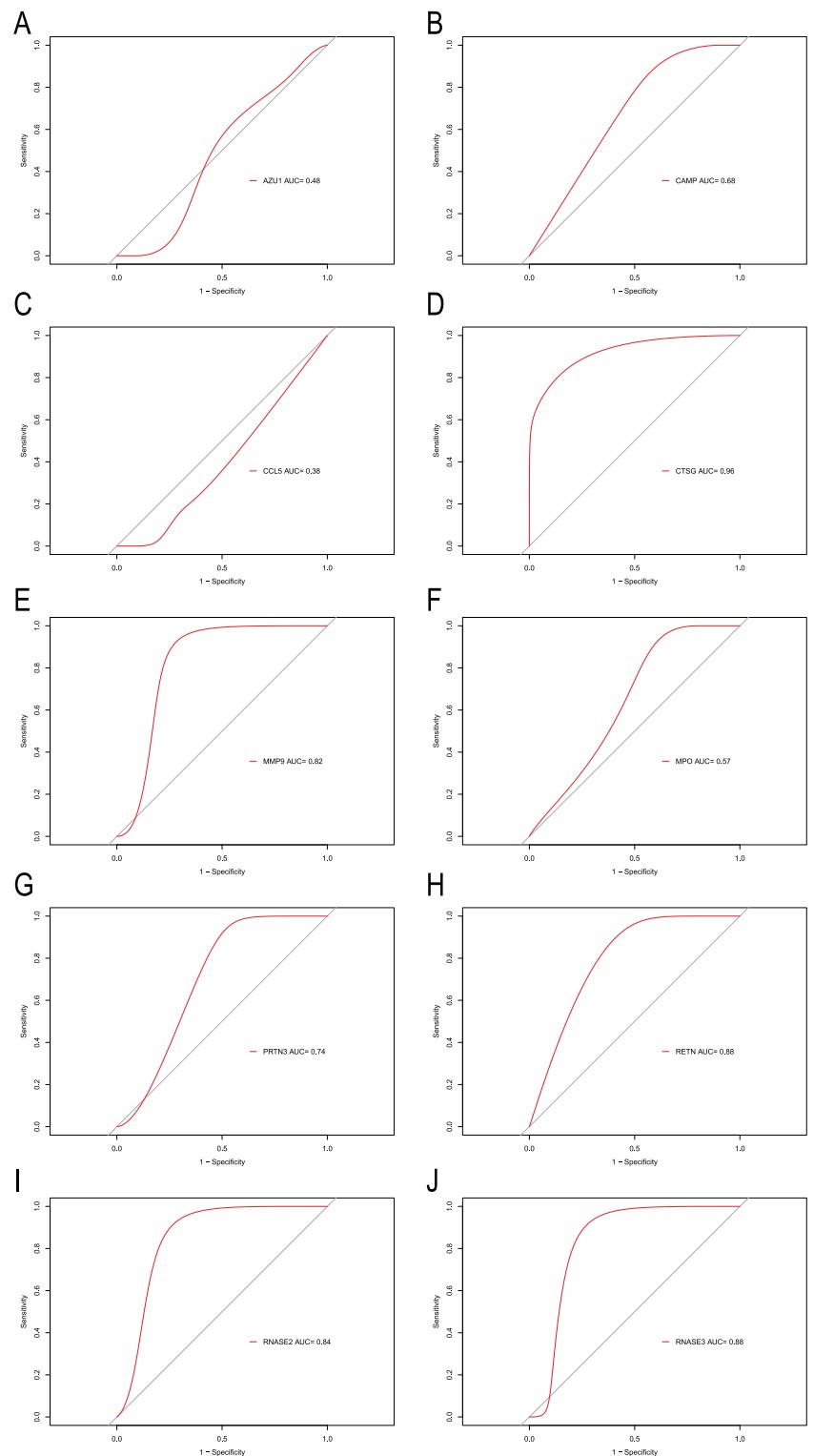

**Figure 6 ROC curves were generated to validate the ability of diagnostic value of the 10 hub genes for CML.** (A) AZU1. (B) CAMP. (C) CCL5. (D) CTSG. (E) MMP9. (F) MPO. (G) PRTN3. (H) RETN. (I) RNASE2. (J) RNASE3.

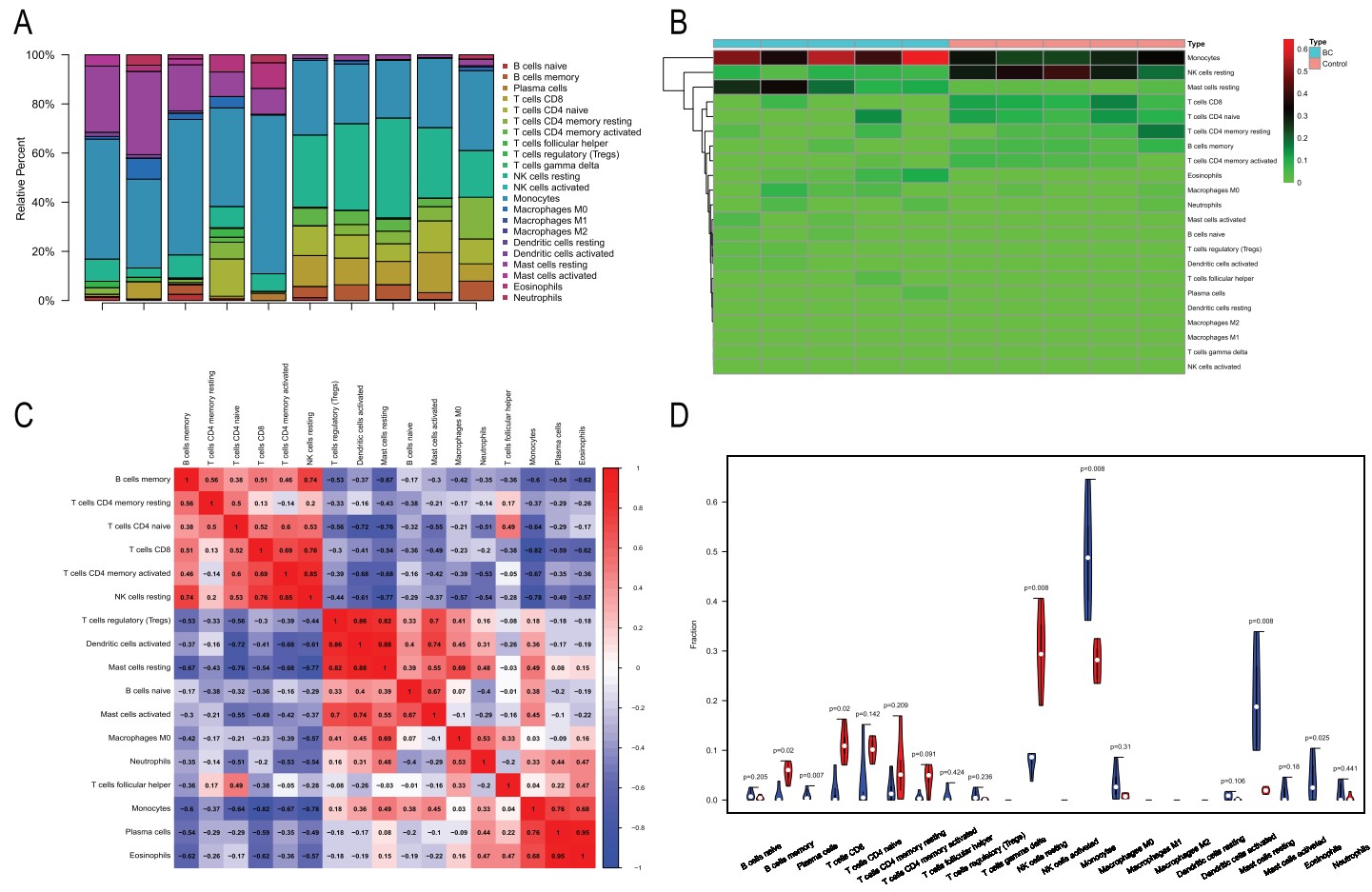

**Figure 7** **Immune infiltration analysis of CML-CP and normal.** (A) A total of 22 Immune cell composition of each sample. (B) Heat map of the 22 immune cell proportions in CML-CP and Normal. (C) The co-expression patterns among fractions of immune cells. (D) The violin graph shows the difference of immune infiltration between CML-CP and Normal.

with the infiltration of CD8⁺T cells, CD4⁺T cells, NK cells resting, macrophages M0, eosinophils (Fig. 12). The results suggest that there is some correlation between the six hub genes and immune response in CML patients.

## Validating hub genes expression in clinical samples

To further assess the *CTSG*, *MMP9*, *PRTN3*, *RETN*, *RNASE2*, and *RNASE3* expression levels in CML, we perform RT-qPCR to search the mRNA expression of the hub genes in PBMCs of the healthy individuals and CML-CP/BP patients (Fig. 13). The results showed that the expression of *CTSG*, *MMP9*, *PRTN3*, *RETN*, *RNASE2*, and *RNASE3* were dramatically upregulated in CML-CP comparing to the normal cells ($P < 0.001$). Compared with CML-CP samples, the expression of *CTSG*, *RETN*, *RNASE2*, and *RNASE3* were significantly lower in the CML-BP samples. The expression trend was consistent with the GSE100026 dataset.

In the present study, the correlation between clinical features of CML and the expression levels of hub genes were analyzed. As shown in Table 3, the expression of *CTSG* was positively associated with white blood cell (WBC) number ($r = 0.4827$, $P = 0.0007$) and

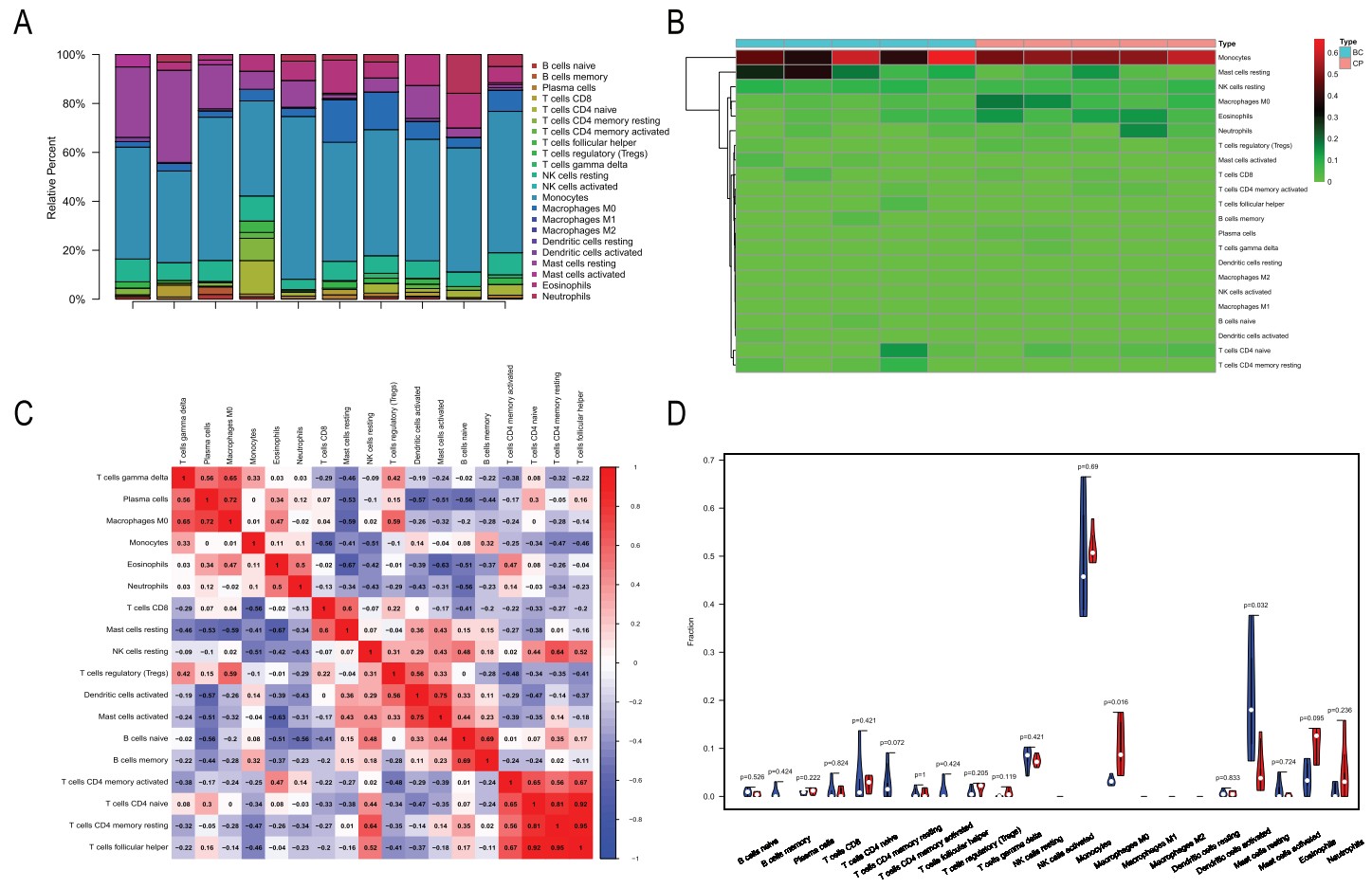

**Figure 8 Immune infiltration analysis of CML-BP and normal.** (A) Immune cell composition of each sample. (B) Heat map of the 22 immune cell proportions in CML-BP and Normal. (C) The co-expression patterns among fractions of immune cells. (D) The violin graph shows the difference of immune infiltration between CML-BP and Normal.

haemoglobin ($r = 0.3135$, $P = 0.0241$). The expression of *PRTN3* was positively associated with haemoglobin ($r = 0.3326$, $P = 0.0128$). The expression of *RETN* was positively associated with WBC number ($r = 0.3832$, $P = 0.0027$). Furthermore, the expression levels of *MMP9*, *RNASE2*, and *RNASE3* in PBMCs from patients with CML did not correlate with WBC number, haemoglobin, and platelet (Table 3).

## DISCUSSION

Approximately 15% of all new leukemia cases are CML according to American Cancer Society estimates (*Apperley, 2015*). The use of TKIs such as imatinib and nilotinib significantly improved the efficacy of CML treatment. TKI treatment over 12 months led 66% of CML patients to have a complete cytogenetic response. Major molecular biological responses were noted in 40% of patients and the 10-year overall survival rate reached 85–90% (*Melo & Barnes, 2007*). However, drug resistance has emerged with diversified and individual characteristics, seriously affecting TKI efficacy (*Deininger, Goldman & Melo, 2000*). Therefore, it is important to develop therapeutic targets aside from ABL kinase.

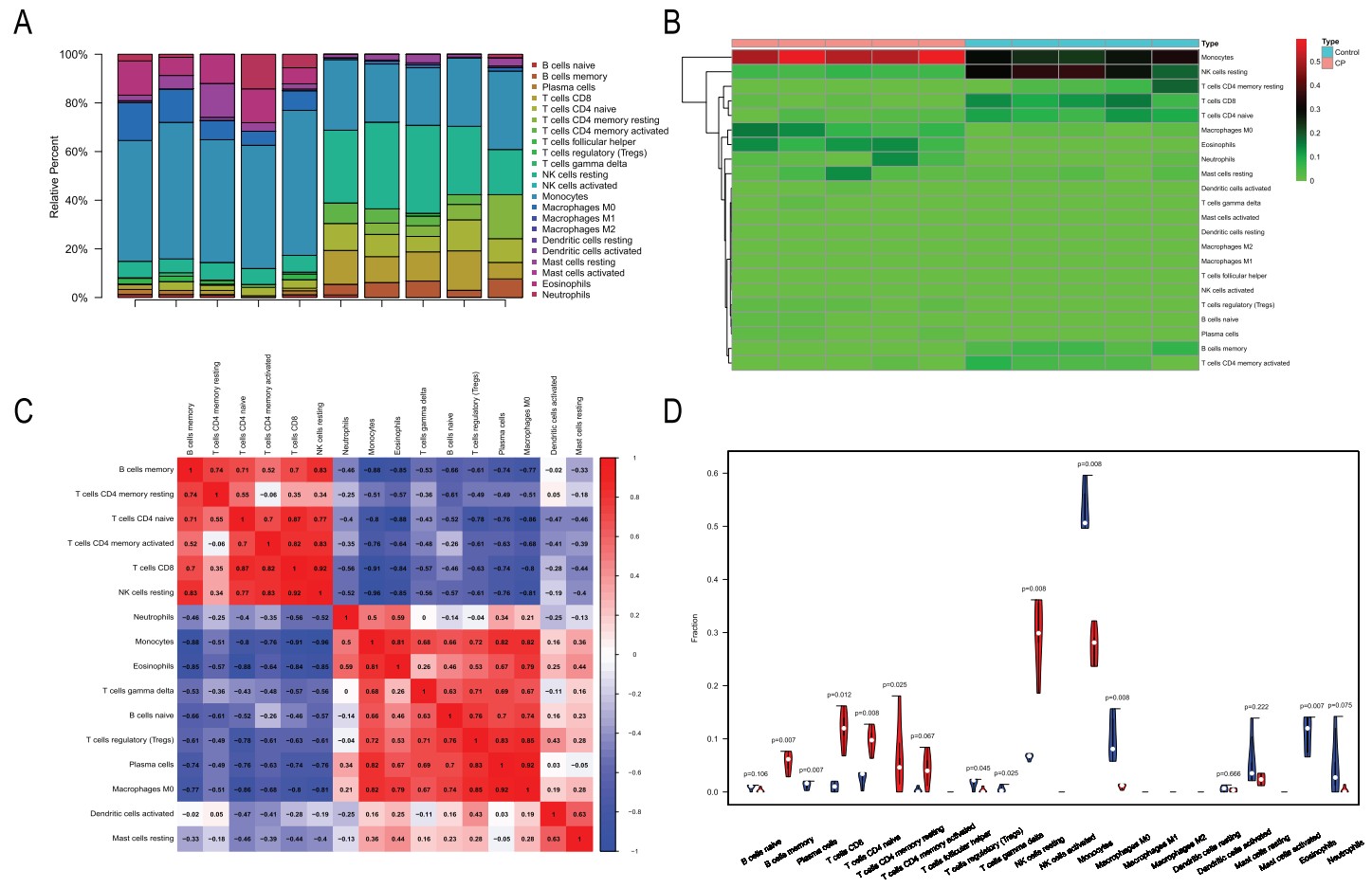

**Figure 9 Immune infiltration analysis of CML-CP and CML-BP.** (A) Immune cell composition of each sample. (B) Heat map of the 22 immune cell proportions in CML-CP and CML-BP. (C) The co-expression patterns among fractions of immune cells. (D) The violin graph shows the difference of immune infiltration between CML-CP and CML-BP.

Recent immunotherapy research has brought to light the relationship between tumors and immunity. The positive response to immunotherapy typically depends on the interaction between tumor cells and immune regulation. Previous studies have investigated the role of immunity in various cancers and have found that immune effector cells such as T and NK cells can effectively eliminate tumor cells (*Sconocchia et al., 2005*). However, immune cells also confer resistance when the oncogenic signaling pathway is activated or the protective mechanism against cell death is active (*Held et al., 2016*). These findings are ready to be applied clinically. For example, the National Institute for Health and Care Excellence (NICE) recommends that some patients using National Healthcare System (NHS) adopt new CAR-T treatments for lymphoma (*Mahase, 2021*). We focused on immune molecule and immune cell changes during CML progression and attempted to locate IRGs to extract key genes and explore effective diagnostic markers. We identified 31 IRGs by comparing the transcriptional expression profiles of CML-CP, CML-BP, and healthy samples. GO and KEGG enrichment analysis revealed that these enriched modules and pathways are closely related to immune cell activation and the immune response

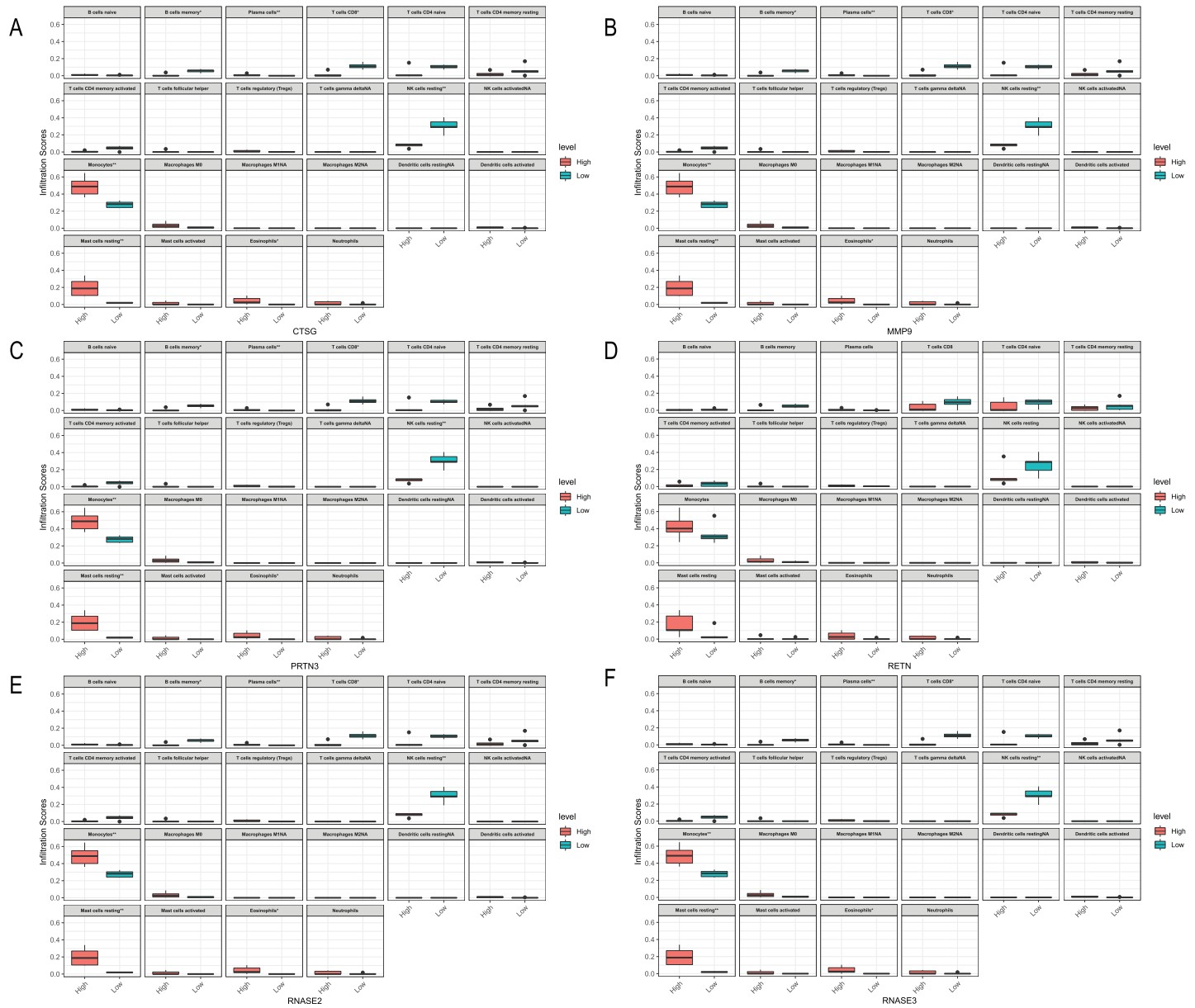

**Figure 10 Relationships between the hub genes and immune cells in CML-CP-normal.** (A) CTSG. (B) MMP9. (C) PRTN3. (D) RETN. (E) RNASE2. (F) RNASE3.

observed in CML. The top 10 hub genes associated with CML were identified in the PPI network. Of these ten, six hub genes (*CTSG*, *MMP9*, *PRTN3*, *RETN*, *RNASE2* and *RNASE3*) may be used as CML diagnostic markers. Their expressions were significantly associated with a variety of immune cell disorders by immune infiltration analysis.

We performed GO enrichment analysis to investigate the 31 IRGs' BPs. Of the MF annotations, receptor ligand activity, cytokine activity, endopeptidase activity were found to be significantly associated with CML occurrence and development. We performed KEGG analysis to determine the biological functions of the 31 immune-related genes associated with CML. Th17 cell differentiation, NF-kappa B signaling pathway, and
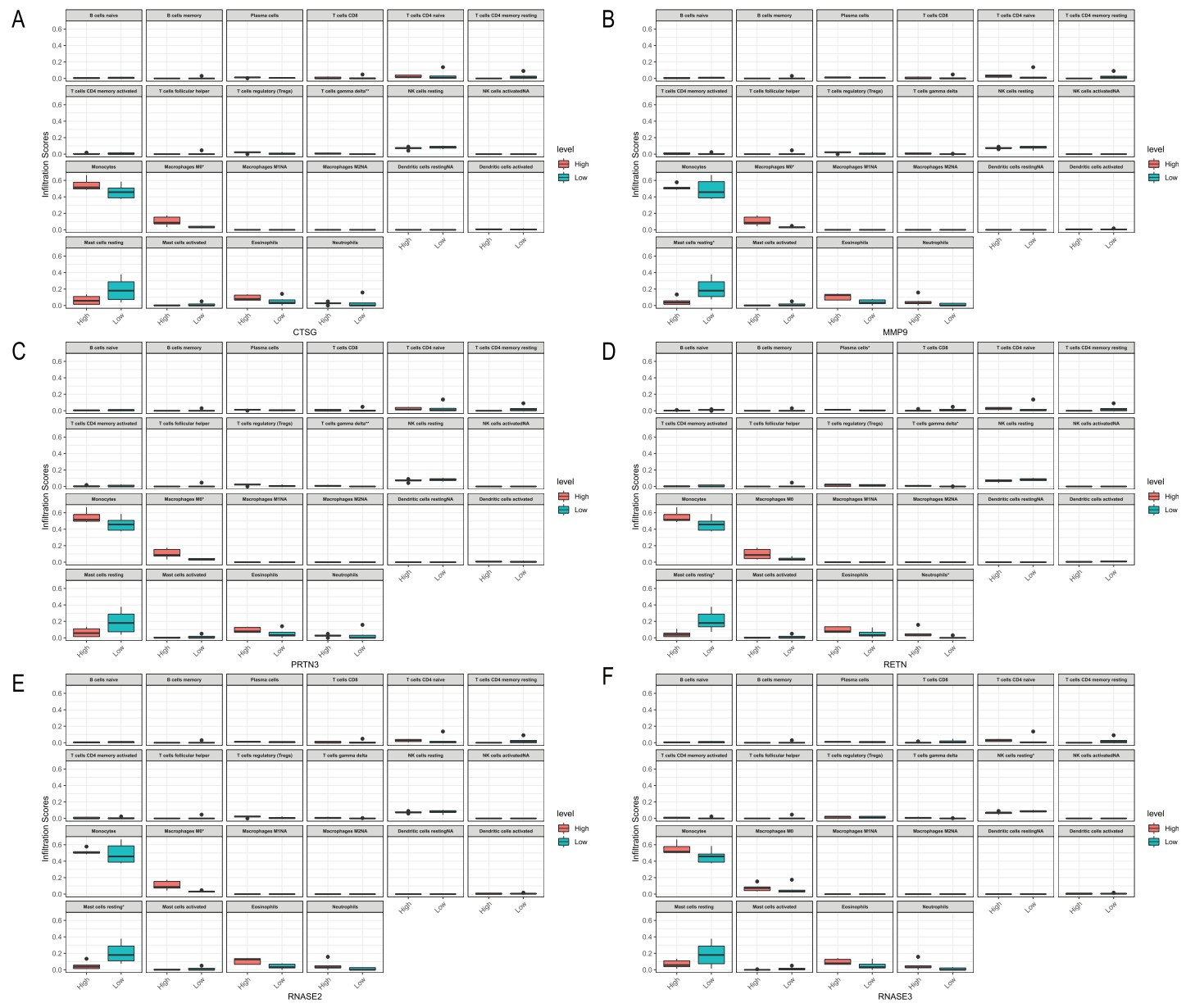

**Figure 11 Relationships between the hub genes and immune cells in CML-BP-normal.** (A) CTSG. (B) MMP9. (C) PRTN3. (D) RETN. (E) RNASE2. (F) RNASE3.

cytokine-cytokine receptor interaction were the three most significantly enriched pathways. Interestingly, some enriched pathways were associated with CML. For instance, NF-κB is activated through the canonical IKK pathway and plays distinct roles in the pathogenesis of myeloid and lymphoid leukemias induced by BCR-ABL1. This confirms that NF-κB and IKKs are targets for Ph(+) leukemias therapy (*Hsieh & Van Etten, 2014*). Previous studies demonstrated that IL-2-activated NK (ANK) with a potent major histocompatibility complex that had unrestricted cytotoxic activity and suppressed malignant hematopoiesis. These observations support the use of autologous ANK therapy to prevent the relapse of CML after autologous marrow transplantation or the use of ANK

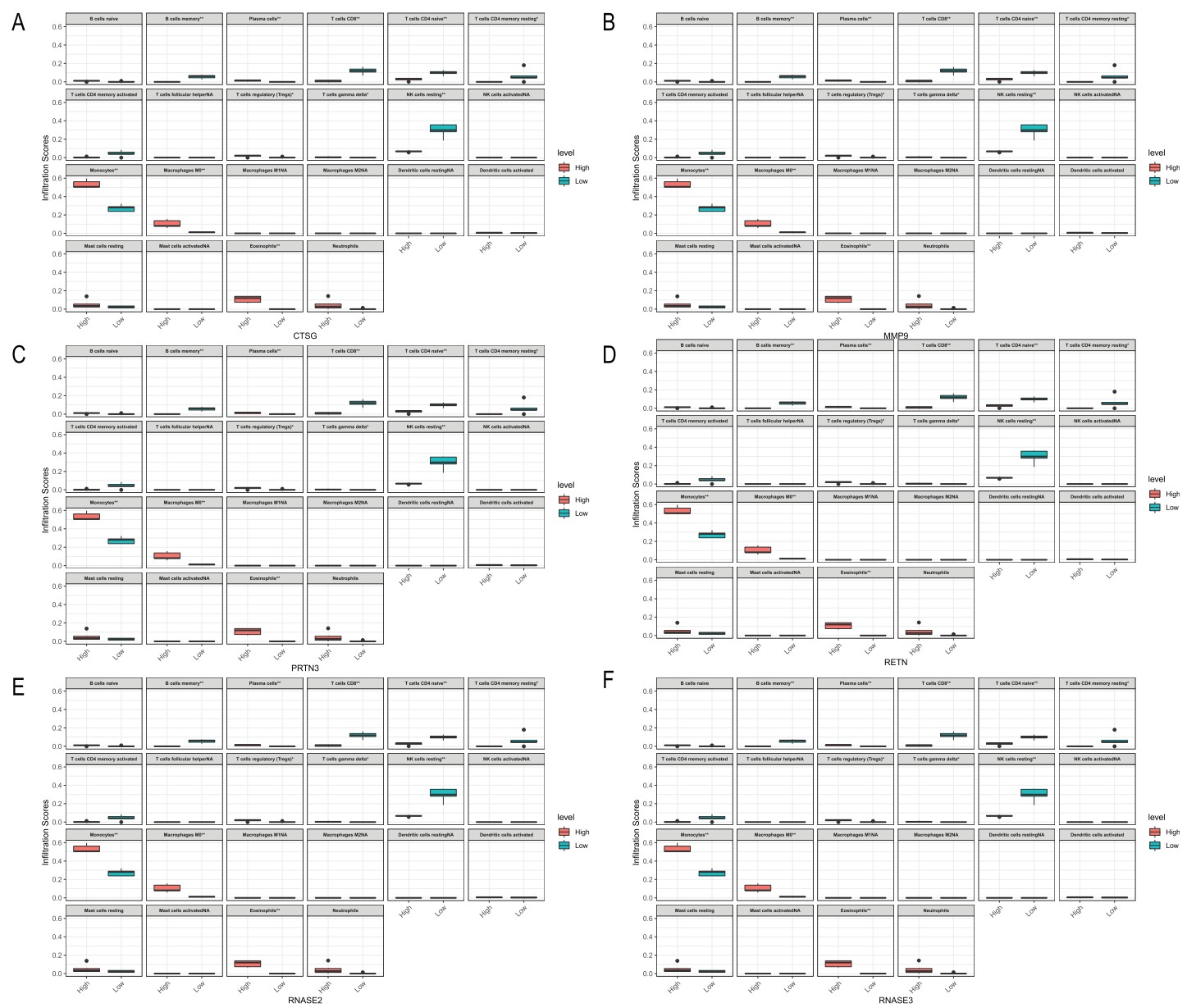

**Figure 12 Relationships between the hub genes and immune cells in CML-CP-CML-BP.** (A) CTSG. (B) MMP9. (C) PRTN3. (D) RETN. (E) RNASE2. (F) RNASE3.

to purge CML marrow for autologous transplantation (*Cervantes et al., 1996*). Autology-activated NK cells can inhibit primitive CML progenitors in long term cultures. CML cells can avoid immune escape *in vivo*, resulting in NK cell-mediated immune destruction (*Cervantes et al., 1996*). IFN-γ released by activated T or NK cells modulate the sensitivity of CML cells to TKI, thereby interfering with the therapeutic effect of TKI on CML. This effect suggests that inflammation-mediated BCR-ABL1 independent resistance mechanisms are significant (*Held et al., 2016*).

In the PPI network identified in the present study, 10 IRGs were determined to be the most significant hub genes, with multiple interactions in the network. The 10 hub genes'

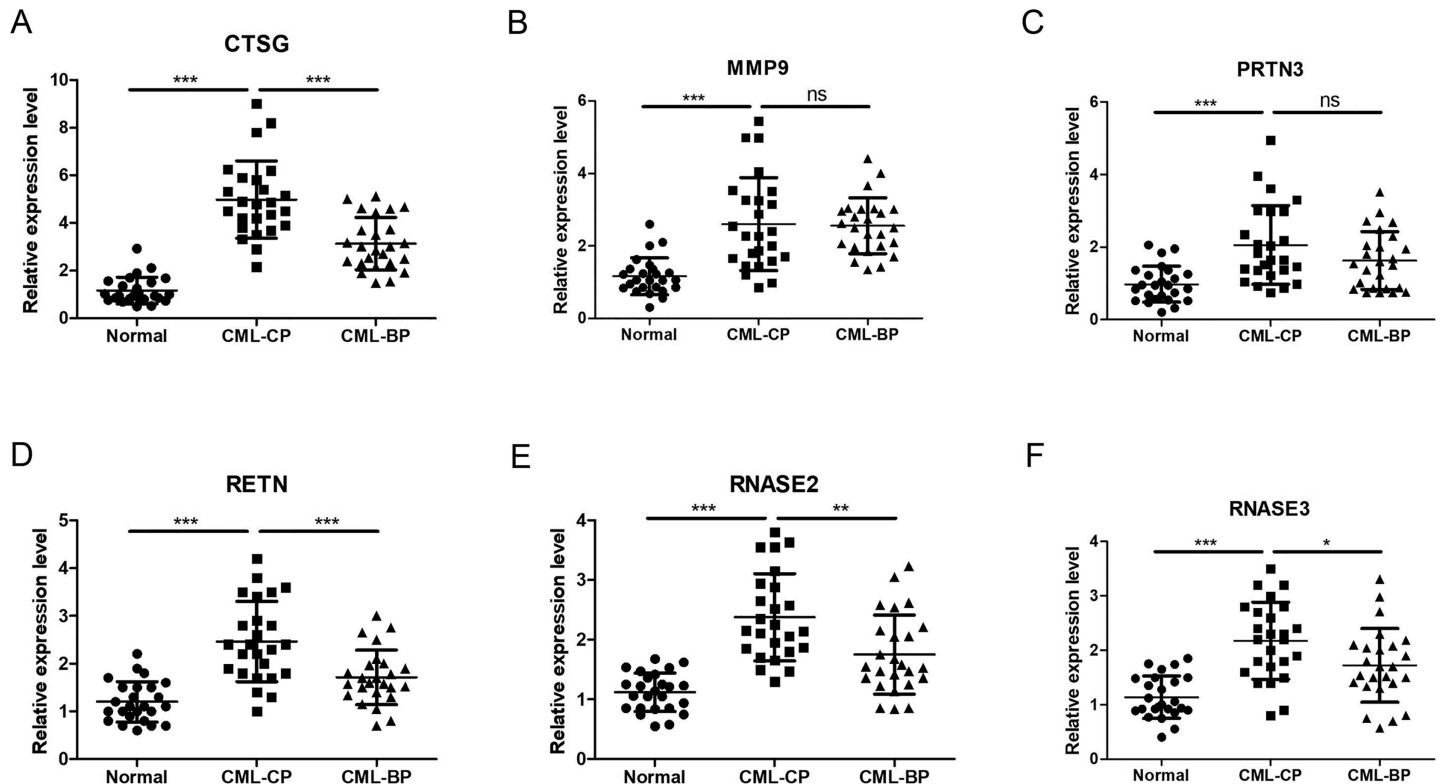

**Figure 13 Related expression levels of hub genes.** (A) CTSG. (B) MMP9. (C) PRTN3. (D) RETN. (E) RNASE2. (F) RNASE3. Significant differences were supposed at $^*P < 0.05$; $^{**}P < 0.01$; $^{***}P < 0.001$.

**Table 3 Correlation of the expression of six hub genes with clinical characteristics of CML.**

| Categories | CTSG | | MMP9 | | PRTN3 | | RETN | | RNASE2 | | RNASE3 | |
|---|---|---|---|---|---|---|---|---|---|---|---|---|
| | r | P-value | r | P-value | r | P-value | r | P-value | r | P-value | r | P-value |
| Age | 0.0328 | 0.8476 | 0.0215 | 0.8041 | 0.0755 | 0.6429 | 0.0952 | 0.5738 | 0.0415 | 0.7928 | 0.0617 | 0.6952 |
| WBC | 0.4827 | 0.0007 | 0.1913 | 0.2267 | 0.1625 | 0.3009 | 0.3832 | 0.0027 | 0.2376 | 0.1443 | 0.1954 | 0.2217 |
| Haemoglobin | 0.3135 | 0.0241 | 0.0952 | 0.6673 | 0.3326 | 0.0128 | 0.1884 | 0.5246 | 0.1268 | 0.4572 | 0.1668 | 0.3058 |
| Platelet | 0.1769 | 0.2745 | 0.1364 | 0.4115 | 0.2328 | 0.1494 | 0.2159 | 0.1776 | 0.1675 | 0.3894 | 0.0364 | 0.8215 |

AUCs were analyzed with regard to diagnostic value. An AUC value greater than 0.7 was the screening criteria. Six hub genes (*CTSG*, *MMP9*, *PRTN3*, *RETN*, *RNASE2*, *RNASE3*) were determined to be diagnostic markers for CML. The AUC values ranged from 0.74–0.96. The RT-qPCR results showed that CTSG, MMP9, PRTN3, RETN, RNASE2, RNASE3 were significantly elevated in PBMCs of CML patients compared with healthy controls. GSEA showed that these hub genes were predominantly associated with the interferon gamma response, inflammatory response, and allograft rejection.

Cathepsin G (CTSG) plays an important role in host defense and can escape intracellular monitoring systems by preventing degradation of foreign proteins. CTSG is broadly expressed in acute myeloid leukemia (AML) blasts and leukemia stem cells (*Alatrash, 2013*). Recent studies have shown that *CTSG* is an effective target for the

immunotherapy of AML and acute lymphoid leukemia (ALL) (*Groborz et al., 2019*). We found that CTSG was highly-expressed in CML-CP and CML-BP. We also found that the AUC of CTSG was 0.96, indicating that it has high diagnostic value. Matrix metalloproteinase 9 (MMP9) upregulation in leukemic cells promotes the blast crisis in CML (*Nakahara et al., 2014*). In chronic lymphocytic leukemia (CLL), MMP9 contributes to CLL pathology by regulating cell survival and migration, promoting angiogenesis, and is associated with poor prognosis (*Aguilera-Montilla et al., 2019*). It has been shown that the ID1 transcriptional inhibitor-MMP9 axis can enhance the invasiveness of BCR-ABL1 transformed leukemia cells (*Nieborowska-Skorska et al., 2006*). Human proteinase 3 (PRTN3) is a leukemia-associated antigen. The identified PRTN3(235)-epitope can be used to study the role of CD4+Th- and Treg-cells in immune responses against PRTN3 in leukemia patients (*Berlin et al., 2015*). Resistin (RETN), a proinflammatory cytokine, is elevated in a number of pathological disorders, including leukemia. The serum RETN level in all patients was elevated and may serve as a potential clinical diagnostic marker to detect the recurrence of leukemia (*El-Baz et al., 2013*). Human eosinophil derived neurotoxin (EDN, RNase2) and eosinophil cationic protein (ECP, RNase3) sequences may have up to a 92% identity in their promoter regions (*Wang et al., 2009*). It produces host defense effects by promoting leukocyte activation, maturation, and chemotaxis. *Niini et al. (2002)* demonstrated that RNase2 was abnormally expressed in childhood acute lymphoblastic leukemia (a cDNA array) and was associated with prognosis. These findings are consistent with our work, which suggests that there may be a close relationship between CML and immune composition.

Our study had some limitations. First, the sample size was small; a larger sample size is needed to verify our results. Second, some clinical data are lacking due to the heterogeneity, long duration, and difficulty of follow-up of CML. It is difficult for us to assess the relationship between risk indicators and patient stratification based on the severity of CML. In order to determine the value of hub genes related to CML as diagnostic and therapeutic biomarkers, more clinical variables should be used for further external verification. Third, the potential mechanism of the six hub genes in CML progression is still not well understood and additional studies may be needed to determine the six hub genes' underlying molecular mechanisms in CML.

## CONCLUSIONS

Ours is the first study to comprehensively identify IRGs in CML. In total, 31 IRGs and 10 hub genes were identified and may serve as novel potential targets for the diagnosis and treatment of CML. However, more molecular experiments are needed to validate our findings.

### Funding

This work was supported by the National Natural Science Foundation of China (No. 81860034) and the Science and Technology Project of Health Commission of Jiangxi

Province (No. 20203755). The funders had no role in study design, data collection and analysis, decision to publish, or preparation of the manuscript.

## Grant Disclosures
The following grant information was disclosed by the authors:
National Natural Science Foundation of China: 81860034.
Science and Technology Project of Health Commission of Jiangxi Province: 20203755.

## Competing Interests
The authors declare that they have no competing interests.

## Author Contributions
- Fangyi Yao conceived and designed the experiments, performed the experiments, analyzed the data, prepared figures and/or tables, authored or reviewed drafts of the paper, and approved the final draft.
- Cui Zhao conceived and designed the experiments, performed the experiments, prepared figures and/or tables, authored or reviewed drafts of the paper, and approved the final draft.
- Fangmin Zhong conceived and designed the experiments, performed the experiments, authored or reviewed drafts of the paper, and approved the final draft.
- Tingyu Qin conceived and designed the experiments, authored or reviewed drafts of the paper, and approved the final draft.
- Shuqi Li conceived and designed the experiments, analyzed the data, authored or reviewed drafts of the paper, and approved the final draft.
- Jing Liu analyzed the data, prepared figures and/or tables, authored or reviewed drafts of the paper, and approved the final draft.
- Bo Huang analyzed the data, prepared figures and/or tables, authored or reviewed drafts of the paper, and approved the final draft.
- Xiaozhong Wang conceived and designed the experiments, performed the experiments, analyzed the data, prepared figures and/or tables, authored or reviewed drafts of the paper, and approved the final draft.

## Data Availability
The raw measurements are available in the Supplemental Files.

## Supplemental Information
Supplemental information for this article can be found online at http://dx.doi.org/10.7717/peerj.12616#supplemental-information.

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
