# Peer review of "Bioinformatics analysis and identification of hub genes and immune-related molecular mechanisms in chronic myeloid leukemia"

_PeerJ, doi:10.7717/peerj.12616_

## Round 0.1 · original submission · Minor Revisions

Dear Dr. Wang: I am delighted to inform you that we will further consider your manuscript upon addressing the concerns/questions by all three reviewers.

Reviewer 1 ·

Basic reporting

no comment

Experimental design

no comment

Validity of the findings

no comment

Additional comments

Manuscript Title: “Bioinformatics analysis and identification of hub genes and immune-related molecular mechanisms in chronic myeloid leukemia”
Summarizing the main findings of the study:
Fangyi Yao al. used the GSE100026 datasets、ImmPort database and MSigDB database to identify the DEGs from Chronic myeloid leukemia (CML). Since, the CML has a significant related to immune, authors aimed to identify immune-related hub genes, pathways, and the expression profiles using an integrative bioinformatics analysis. Also, authors have utilized various bioinformatic pipelines to analyse the datasets such as R software package, Cytoscape, KEGG. DEGs were separately analyzed by Gene Ontology (GO) enrichment and Kyoto Encyclopedia of Genes and Genomes (KEGG) with Metascape database, the protein–protein interaction (PPI) network of the DEGs was analyzed by STRING and visualized in Cytoscape, 10 key genes from PPI network by MCODE plug-ins. Furthmore ,GSEA to get the related pathway and CIBERSORT is used to evaluate the composition of immune cells and so on . in the end of the aticle, the writer clollected serum of patients to test the hub gene expression by qRT-PCR.

I would recommend the manuscript to be accepted with minor revisions.
Minor concerns:
1. Firstly, authors are requested to provide the workflow .using brief word to describe your article structure in the introduction.
2. The figure 9 is unclearly that can not explian the result,please review .
3. The method is not be provied about “Correlation analysis hub gene “. Please review it and explian how to get annlysis hub gene expression of corralation in three databases.
4. Please provied the patients clinicopathological.and get the correlation between the hub gene expression and clinicopathologica.

Reviewer 2 ·

Basic reporting

1. In line 53 and 54, the authors cited only 2021 references in their introduction of the features of CML. However, these features of CML should have been discovered earlier to 2021. Please citing the references correctly and to check other references in this study.

2. The full name was not provided when the abbreviation first appeared, such as line 30, 43 and 44. It is recommended to modify and review the full text again.

Experimental design

1. In the part of Validating hub genes expression in clinical samples (line 210), the authors only provided the q-PCR results of the healthy donors and CML patients, but did not provide the relevant clinical information, such as the age and sex of the healthy donors and CML patients, the routine blood test and diagnosis of the CML patients. Because age, gender and presence of infection may affect the expression of immune-related genes. So, I think it is necessary for the authors to provide these information and detail the age and sex matching between healthy donors and CML patients.

2. The description of the part of GESA analysis of hub genes (line 186), Hub genes for CML diagnosis (line 194), Immune infiltration analysis (line 199), Correlation analysis between diagnostic markers and immune cells (line 204) and Validating hub genes expression in clinical samples (line 210) were too simple that the authors only showed the figures. It is better to add a detailed description of the results, in order to facilitate more readers’ understanding of this study.

3. In line 95, the authors find the IRGs by using the intersection of the differentially expressed genes and the immune genes. What is the immune genes? A database or others research?

4. In line 110, the authors retained the genes with an AUC value greater than 0.7. Why selected 0.7 and is it supported by the literature?

Validity of the findings

1. In line 209, “This indicated that immunity was poor.” But I could not find more description or evidence to support this conclusion. So, I suggest the authors to add a detailed description to illustrate it clearly.

Reviewer 3 ·

Basic reporting

This manuscript is well-written in general.

1) The writings on the plots of Figure 3b and Figure 3d are not visible. It is better to increase the text size.
2) “et al.” should be written in italic.
3) The primer sequences under the heading “RNA isolation and quantitative real-time PCR (RT-qPCR)” (lines 144 - 153) are better to be given as a table.
4) Gene names should be written in italic (lines 183 - 184, 205 - 206, 211).
5) It is better to use “the efficacy of CML treatment” instead of “the efficacy of treatment of CML” (line 219).
6) The sentence in lines 241 – 242 should be changed as “Their expressions were significantly associated with a variety of immune cell disorders by immune infiltration analysis.”.
7) “in vivo.” should be written italic (line 258).
8) BCR-ABL1 is also mentioned as BCR/ABL or BCR-ABL in different paragraphs of the manuscript. They all should be corrected as BCR-ABL1.

Experimental design

It is regarding CML is informative for the readers. Experiments are well defined and the results are meaningful. Bioinformatics analysis is dealt with in detail.

Validity of the findings

Even though the patient samples were limited, the findings seem valuable. If supported with further experiments novel marker or therapeutic target candidates may be defined for CML. The paper is scientifically adequate.

---

## Round 0.2 · Minor Revisions

Please address Reviewer 1's minor points.

Reviewer 2 ·

Basic reporting

No comment

Experimental design

No comment

Validity of the findings

No comment

Additional comments

In line 236-237, “ T cellscells CD8, and T cells CD4 naive” The description of T cell subsets does not seem to conform to the conventional description. It is generally described as CD8+ T cells, CD4+ T cells. So, I suggested that the authors revise it and check the full manuscript.

Reviewer 3 ·

Basic reporting

The revision looks good.

Experimental design

The revision looks good.

Validity of the findings

The revision looks good.

---

## Round 0.3 · accepted · Accept

Dear Dr. Yao,

We are happy to inform you that your manuscript has been Accepted for publication. Congratulations!